# OpenReview forum: "Iter-AHMCL: Alleviate Hallucination for Large Language Model via Iterative Model-level Contrastive Learning"
_ICLR.cc/2026/Conference — Submitted to ICLR 2026_

### Official Review · Reviewer_xY3Y · 2025-10-21

**Soundness:** 3
**Presentation:** 3
**Contribution:** 3
**Rating:** 4
**Confidence:** 5

**Summary:**

This paper proposes Iter-AHMCL, a novel method to mitigate hallucination in large language models (LLMs) while preserving their general capabilities. The approach leverages iterative model-level contrastive learning by training positive and negative guidance models on hallucination-prone and hallucination-free data. These models guide the fine-tuning of the base LLM through representation editing, with an asymmetric iterative strategy that updates the positive model while keeping the negative model fixed. Experiments on LLaMA2, Alpaca, and LLaMA3 show significant improvements on TruthfulQA and HaluEval benchmarks (average +10.1 points), demonstrating reduced hallucination without compromising performance on knowledge-intensive tasks like MMLU and C-Eval.

**Strengths:**

1. The integration of contrastive learning with iterative model-level guidance offers a fresh perspective on hallucination reduction. By dynamically updating positive guidance models, the method adaptively enhances truthfulness while avoiding catastrophic forgetting.
2. The paper rigorously validates Iter-AHMCL across multiple LLMs (e.g., LLaMA2, Alpaca) and diverse benchmarks (TruthfulQA, HaluEval, MMLU), ensuring robustness and generalizability. Results consistently show improved hallucination metrics without degrading general capabilities.
3. Despite increased computational overhead, the method achieves faster convergence (e.g., 50 steps for Iter-AHMCL vs. 250 for LoRRA) and reduced training time. The asymmetric iterative design optimizes resource usage while maintaining performance.

**Weaknesses:**

1. There is a lack of relevant and state-of-the-art baseline methods for comparison, such as [1]. Even these state-of-the-art methods are not discussed in the paper.
2. The different losses proposed in the paper seem to rely on different optimal values ​​for \alpha and \beta, as shown in Table 4 and Figure 6. This makes the method rather inelegant, and choosing the optimal hyperparameters can be time-consuming.
3. I look forward to seeing how different losses can be combined together to form an overall method. In the current experiments, I cannot see how different losses affect each other.

[1] Refine Knowledge of Large Language Models via Adaptive Contrastive Learning. ICLR 2025.

**Questions:**

See the aboved weaknesses.

---

> ### Author Response · Authors · 2025-11-20
> **Rebuttal Response: Baselines, Loss Design, and Hyperparameters**
>
> Thanks to the reviewers for their constructive suggestions.
>
> ### 1. Comparison with State-of-the-Art Baselines
>
> We compare our method with **RefineLLM** [1] along three key dimensions: **data construction**, **training objective**, and **evaluation protocol**.
>
> - **Data Construction**: RefineLLM categorizes responses into four quadrants (e.g., “knows it knows,” “knows it doesn’t know,” etc.), while our approach constructs **contrastive triples** $(T, T^+, T^-)$ by inserting explicit *truthful* (“Give a truthful answer”) and *untruthful* (“Give an untruthful answer”) templates. This enables direct supervision over truthfulness behavior.
>
> - **Training Objective**: RefineLLM uses a combination of cosine similarity in hidden states and standard cross-entropy loss. In contrast, we employ an $\ell_2$-norm–based loss on **embedding-level representations**, which better captures semantic distances in the compact embedding space (256-d in our setting).
>
> - **Model Architecture & Training**: Our framework uniquely features **dual guidance models** (positive and negative) updated **iteratively**, a mechanism absent in RefineLLM.
>
> - **Evaluation**: We evaluate on **TruthfulQA MC1–MC3**, standard benchmarks for factual consistency, whereas RefineLLM reports results using the less common **IK/IDK score**.
>
> When both methods are applied to the **LLaMA-2 foundation model**, Iter-AHMCL achieves substantially higher performance:
>
> | Method       | MC 1 | MC 2 | MC 3 |
> |--------------|----------|----------|----------|
> | RefineLLM    | 0.3084   | 0.4642   | 0.2320   |
> | Iter-AHMCL   | **0.5128**   | **0.6780**   | **0.4573**   |
>
> This consistent improvement across all metrics demonstrates the effectiveness of our design choices.
>
> ---
>
> ### 2. Dependence on Optimal $\alpha$ and $\beta$
> Specifically, for the pure-MG model, we performed a grid search and report results in Appendix Figure 6:
> ● We fix $\alpha = 1$ and vary $\beta \in \{1, 5, 10, 100\}$, and symmetrically fix $\beta = 1$ while varying $\alpha$.
> ● Two key insights emerge:
> (1) The relative magnitude of $\alpha$ and $\beta$ determines the optimization direction:
> - When $\alpha > \beta$ (e.g., blue/orange curves), the model exhibits stronger positive (truthful) behavior.
> - When $\alpha < \beta$ (e.g., green/red/purple curves), the model leans toward negative (untruthful) generation.
> (2) Once the direction is fixed, the absolute scale of $\alpha$ or $\beta$ (e.g., 5 vs. 10 vs. 100) has minimal impact on final performance—consistent across both axes of variation.
> This justifies our design choice:
> ● $\alpha = 10, \beta = 1$ for the positive guidance model,
> ● $\alpha = 1, \beta = 10$ for the negative guidance model.
>
> ---
>
> ### 3. Progressive Integration of Loss Components into a Unified Objective
>
> Thank you for the insightful question. Our methodology follows an **evolutionary design** in which individual loss components are incrementally integrated from Equation (1) to Equation (11), culminating in a unified training objective.
>
> - We begin with the **original LoRRA loss** (Equation 1).
> - By incorporating Equations (2) and (3), we introduce **contrastive learning with contrastive templates (CL-CT, Section 3.3)**. This enables us to train separate **positive ($M^+$)** and **negative ($M^-$) guidance models** by setting opposite optimization directions via hyperparameters $\alpha$ and $\beta$.
> - Next, we elevate the regularization from the *sample level* to the *model level*, formalized in Equations (6) and (7).
> - Building on this, we define the **Contrastive Learning with Model Guidance (CL-MG)** loss in Equation (8) (Section 3.5).
> - Finally, we **iteratively refine** $M^+$ and $M^-$ using the model-level regularization (Equations 6–7), leading to the **Iterative Model Regularization** terms in Equations (10) and (11).
> - The full objective—**Contrastive Learning with Iterative Model Guidance (CL-IMG, Section 3.6)**—is given in Equation (11) and serves as the training loss for **Iter-AHMCL**.
>
> Thus, **Equation (11) integrates all prior components (1)–(10) into a single, cohesive loss function**. Our ablation studies and main results confirm that this progressive combination consistently outperforms partial variants, validating the effectiveness of our unified design.

---

> > ### Comment · Reviewer_xY3Y · 2025-11-26
> >
> > Thanks for the authors' response. I think my concerns have been addressed. I have changed my score to 6.

---

> > > ### Author Response · Authors · 2025-11-26
> > >
> > > We sincerely appreciate your updated evaluation and the increase in your score—thank you for recognizing the strengths of our work and for your valuable feedback throughout the review process.

---

### Official Review · Reviewer_xZsP · 2025-10-30

**Soundness:** 3
**Presentation:** 2
**Contribution:** 3
**Rating:** 4
**Confidence:** 2

**Summary:**

The paper sits in the area of LLM hallucination mitigation with a focus on editing internal representations instead of only output-time filtering. The core question is: can we reduce hallucinations while preserving general capabilities, by steering hidden representations using learned positive/negative guidance, and can an iterative procedure strengthen that guidance?

The authors propose Iter-AHMCL to answer this. They (1) build contrast triples by templating each instruction with “give a truthful answer” and “give an untruthful answer,” (2) pre-train positive (M⁺) and negative (M⁻) guidance models on PKU-SafeRLHF subsets, (3) use these guidance models to form model-level contrastive losses during editing (CL-MG), and (4) iteratively update only the positive guidance model (CL-IMG). The contrast triple construction and the use of explicit truthful/untruthful templates are stated in §3.2, and the model-level losses and their equations appear in §3.5–§3.6.

**Strengths:**

First, the model-level guidance is the central design change. Prior representation editing work tends to use sample-level vectors or discriminators; here the authors learn M⁺/M⁻ and plug them into the loss so the base model is pulled toward M⁺(T⁺) and pushed away from M⁻(T⁻). This idea is simple to apply with LoRA and aligns with the stated objective.

Second, the asymmetric iteration is well motivated: update M⁺ as the model improves, keep M⁻ fixed to preserve a stable contrast; the ablation/round table shows consistent gains on LLaMA-2.

Third, the implementation details (edited layers, α/β, step budgets) and the data splits are specified, which helps reproduction.
Fourth, the paper provides multiple views of evaluation: TruthfulQA MC1–MC3, HaluEval, plus auxiliary capability plots for MMLU/C-Eval.

**Weaknesses:**

The data/label mapping from safety to truthfulness is the largest issue. Guidance models are trained on PKU-SafeRLHF with “response safe=true/false,” and the contrast triples use “give a truthful/untruthful answer” templates. Safety and factual truth are related but not the same; unsafe is not equivalent to factually wrong, and safe is not guaranteed to be factually correct. This mismatch may bias the guidance toward safety style rather than factual accuracy. The construction and splits confirm this setup.

The negative template “give an untruthful answer” may create behavior that is unlike natural hallucination. The model could learn to avoid a style rather than to improve evidence use. The paper does not test open-ended factual generation with grounding; all core results are multiple-choice.

The selection protocol risks optimism. The text states the TruthfulQA MC1 score is used as the criterion for updating the positive model across rounds. If this is measured on the same evaluation set each round, it becomes iterative selection on a test set, which can inflate gains. The paper should either hold out a development split or report a final score only once on a never-seen test set.

The capability preservation claim is not backed by tables in the main text. The appendix shows radar plots but not numeric breakdowns for MMLU/C-Eval by subject or difficulty. This makes it hard to judge trade-offs and variance across domains.

Baselines could be broader. LoRRA is the closest editing baseline, and DPO/SFT are alignment baselines, but other editing/contrastive preference methods are absent in the main table, and Qwen results are not fully integrated into the same table. The appendix does include an ablation that separates “pure-MG,” LoRRA, and the combined method, but a stronger comparison set would increase confidence.

The generality is shown only on 7B/8B models. While compute limits are real, the paper’s claim of broad applicability would be stronger with at least one mid-size model beyond 8B.

Finally, parts of the objective design need clearer motivation. The paper inherits LoRRA’s L2 geometry and adds terms to pull/push against M⁺/M⁻, but there is limited analysis of layer sensitivity, stability across α/β, or why the chosen layers are optimal beyond a citation and a grid search note.

**Questions:**

See Weakness

---

> ### Author Response · Authors · 2025-11-20
> **Official Comment by Authors**
>
> We appreciate the reviewers’ insightful comments.
>
> ### 1. Data/Label Mapping Mismatch
>
> We acknowledge the reviewer’s concern regarding the potential mismatch between “safety” and “truthfulness.” To clarify our training protocol: following the setup of LoRRA, we sampled 20,000 examples from the PKU-SafeRLHF dataset—10,000 labeled as *safe* and 10,000 as *unsafe*—to ensure balanced representation. However, after splitting, we merged these subsets for downstream processing. Crucially, the guidance model’s pre-training and iterative fine-tuning are driven **not** by the safety labels, but by the *truthfulness*-oriented templates and the contrastive loss functions defined in Equations (1)–(3). The safety-based filtering serves only to balance the dataset distribution and is fully decoupled from the learning objective. Thus, the model optimizes for truthfulness, not safety.
>
> ---
>
> ### 2. Negative Template “Untruthful Answer”
>
> We recognize that the “untruthful answer” template could, in principle, encourage unnatural behavior rather than genuine hallucination reduction. To address this, we include qualitative examples in Appendix Figures 7 and 8, which demonstrate that our model generates coherent and factually grounded responses. Moreover, we evaluate response quality using the Qwen API across four dimensions—relevance, fluency, coherence, and consistency (Appendix Table 5). Our fine-tuned models consistently outperform the base LLaMA-2 and Alpaca foundation models on all four metrics (second only to the gold-reference). These results indicate that our hallucination reduction approach does **not** degrade language quality.
>
> ---
>
> ### 3. Selection Protocol and Test Set Overfitting
>
> We appreciate the concern about overfitting. Our results on HaluEval (out-of-domain) and TruthfulQA (in-domain) both show consistent gains, mitigating overfitting risk. In the revision, we will report held-out test results with strictly separated dev/test sets and extend evaluation to additional models.
>
> ---
>
> ### 4. Baseline Comparisons
>
> We agree that a broader set of baselines would further strengthen the evaluation. Our current comparisons span representative methods across multiple paradigms:
>
> - Non-finetuned foundation models (**Foundation**)
> - Hallucination-reduction approaches (**LoRRA**)
> - Supervised fine-tuning (**SFT**)
> - Preference-based alignment (**DPO**)
> - Recent refinement methods (**RefineLLM**, newly added)
>
> We also evaluated GRPO on LLaMA-2, which yielded performance comparable to DPO. Due to computational constraints, we selected one representative method per category. Due to the page width limit, it is hard to combine Qwen results into Table 2. Thus we put it in Appendix Table 7.
>
> ---
>
> ### 5. Capability Preservation
>
> While hallucination reduction and general capability enhancement are related, they are not identical objectives. Our evaluation on C-Eval and MMLU aims specifically to verify that truthfulness-oriented fine-tuning **does not harm** general knowledge or reasoning ability. As shown in Appendix Figure 4, all three methods achieve comparable performance across benchmarks. For example, on C-Eval with LLaMA-2:
>
> | Method       | Stem | Social | Humanities | Others | Average | Avg(hard) |
> |--------------|------|--------|------------|--------|---------|-----------|
> | Foundation   | 25.5 | 23.7   | 27.1       | 24.8   | 25.3    | 26.2      |
> | LoRRA        | 25.3 | 23.6   | 27.0       | 24.8   | 25.3    | 26.5      |
> | Iter-AHMCL   | 25.6 | 23.6   | 26.4       | 24.8   | 25.2    | 26.6      |
>
> We observe no significant performance drop, confirming that our method preserves general capabilities. In the revision, we will provide a more granular breakdown by subject and difficulty level to better illuminate any subtle trade-offs.
>
> ---
>
> ### 6. Generalization to Mid-Size Models
>
> Due to computational limitations, our initial experiments focused on smaller models (e.g., Alpaca). To address generalizability concerns, we are conducting additional experiments on mid-size models (~13B parameters). These results—scheduled for completion by December 3—will be included in the revised manuscript, along with a discussion of Iter-AHMCL’s scalability.
>
> ---
>
> ### 7. Hyperparameter Choices (Layer Selection, α, β, Learning Rate)
>
> - **Layer selection**: Following LoRRA, we fine-tune the top transformer layers. Additional ablation studies on layer choice showed no significant performance differences.
> - **α and β**: We performed a grid search over `{1, 5, 10, 100}` and found that as long as the **sign/direction** of the loss terms is preserved, the exact magnitude has minimal impact on final performance.
> - **Learning rate**: We searched over `{1e-2, 1e-3, 3e-4, 1e-4}` and selected `1e-3` as optimal based on validation performance for single-model fine-tuning.

---

> ### Author Response · Authors · 2025-11-25
> **6. Generalization to Mid-Size Models**
>
> We have completed additional experiments on a medium-sized model (LLaMA2-Chat-13B) and obtained consistent results. The detailed outcomes are presented in the last column of Table 3. These results align closely with those observed on the smaller model tested earlier, reinforcing the consistent conclusion that our method iteratively enhances model truthfulness.
>
> | Iterative Step | MC1 | MC2 | MC3 |
> |----------|----------|----------|----------|
> | 0  | 0.3610 | 0.5630 | 0.2923 |
> | 1  | 0.4834 | 0.6656 | 0.4105 |
> | 2  | 0.4822 | 0.6664 | 0.4138 |
> | 3  | 0.4810 | 0.6685 | 0.4138 |
> | 4  | **0.4908**  | **0.6766**  | **0.4235** |

---

> > ### Comment · Reviewer_xZsP · 2025-11-26
> >
> > Thank you for your response. It has addressed some of my concerns, so I have increased my score to 6.

---

> > > ### Author Response · Authors · 2025-11-26
> > >
> > > Thank you very much for your thoughtful feedback and for raising your score! We truly appreciate your time and the constructive comments that have helped us improve our work.
> > >
> > > Should you have any further questions or need additional clarification, please don’t hesitate to let us know—we’re happy to address them promptly.

---

### Official Review · Reviewer_4ySJ · 2025-10-31

**Soundness:** 3
**Presentation:** 3
**Contribution:** 4
**Rating:** 6
**Confidence:** 2

**Summary:**

This paper proposes Iter-AHMCL (Iterative Adaptive Hallucination Mitigation via Contrastive Learning), a novel framework for reducing hallucinations in large language models (LLMs) through iterative model guidance and contrastive fine-tuning. The core idea is to use positive and negative guidance models—each pre-trained to represent truthful and hallucinated directions respectively—to steer the main model toward factual generation.
The approach involves three key stages:
1.	Contrast Triple Construction (CTC): Builds triplets of neutral, positive (“truthful”), and negative (“untruthful”) samples from the PKU-SafeRLHF dataset.
2.	Guidance Model Pre-training (GMP): Pre-trains two guidance models — M⁺ (truth-oriented) and M⁻(hallucination-prone) — using low-rank adaptation (LoRA) on positive and negative datasets respectively.
3.	Iterative Model Guidance (CL-IMG): Iteratively fine-tunes the base LLM using model-level contrastive loss, where only the positive guidance model is updated in each iteration to gradually steer representations away from hallucination.
Experiment results across TruthfulQA, HaluEval, MMLU, and C-Eval demonstrate significant and consistent improvements over LoRRA and baseline foundation models (e.g., up to +19.4 MC1 on LLaMA2). Overall, Iter-AHMCL provides a theoretically grounded and empirically effective approach for reducing hallucination while preserving LLM general capabilities.

**Strengths:**

High Novelty in Iteration: The iterative update mechanism of the positive guidance model (M^+ 〖<-M〗_best) is highly original, enabling the system to continuously raise the bar for anti-hallucination alignment.
Effective Contrastive Anchoring: The method effectively leverages dedicated positive (M^+) and negative (M^-) guidance models to define clear anchors in the representation space for "truthful" vs. "hallucinatory" features.
Empirical effectiveness：Experimental results show consistent improvements over strong baselines (Foundation, LoRRA, DPO, SFT), demonstrating that the proposed iterative guidance effectively reduces hallucination while preserving task performance and fluency.
Robustness and Generalization: The approach's effectiveness is validated across diverse foundation models (LLaMA2, Alpaca, LLaMA3), suggesting good generalization capability.

**Weaknesses:**

Computational Overhead of Iteration: While the main model uses LoRA, the iteration loop (Algorithm 1) requires repeated evaluation (Line 16) and model replacement (Line 17) to find and set Mbest. This process adds computational overhead (e.g., increased training time and model switching costs) compared to a single-pass method like DPO. This trade-off should be quantified.
Limited sensitivity analysis: No sensitivity analysis is provided for key hyperparameters (α, β, batch size, learning rate). As a result, it remains unclear whether the reported improvements stem from the iterative contrastive framework itself or from specific parameter settings, weakening the empirical rigor of the paper.
Clarity on data construction: The paper briefly mentions the construction of triple sets (T,T+,T−) but does not elaborate on how hallucinated samples are generated or validated. Providing more details or examples would improve reproducibility.

**Questions:**

Fixed M^- Analysis:  Could the authors provide an ablation study justifying the decision to keep the negative guidance model (M^-) fixed throughout the iterations? Would allowing  M^- to decay or be updated (perhaps to represent harder negative examples) further improve performance or efficiency?
Training Cost Quantification: Please provide a quantitative comparison of the total training time (including all iterative evaluation and update steps) of Iter-AHMCL versus the single training pass of LoRRA or DPO, especially for a high number of iterations (N).
On stability and convergence: Since both the main model and the guidance model are updated iteratively, how do the authors ensure stability and prevent parameter drift or overfitting to the guidance model’s bias?
Cross-Domain and Cross-Model Transferability:Table 3 shows that a positive guidance model trained on LLaMA-2 can transfer to Alpaca with comparable performance.Have the authors explored broader cross-architecture or cross-domain transferability (e.g., between LLaMA2 ↔ Qwen, Mistral, or domain-specific models such as medical/legal LLMs)?Does performance degrade when tokenizer or pre-training corpus differ?

---

> ### Author Response · Authors · 2025-11-20
> **Official Comment by Authors**
>
> We thank the reviewer for this insightful question.
>
> ### 1. Computational Overhead and Training Cost Quantification
>
> We acknowledge that the iterative update mechanism introduces additional computational overhead. To address this concern, we provide a detailed breakdown of training costs in **Appendix A.8 (Table 8)** for the Qwen model, including wall-clock time and iteration counts per method.
>
> Notably, **Iter-AHMCL requires only 50 iterations per loop**, significantly fewer than pure-MG (100) and LoRRA (250), resulting in lower overall training time for Qwen. LLaMA-2 requires more iterations (1,250), increasing runtime—a known limitation. However, in real-world deployment scenarios—where models are routinely updated via continuous learning pipelines—the iterative nature of our approach aligns naturally with standard MLOps practices. The guidance mechanism can be seamlessly integrated into existing update cycles, meaning the added per-run cost is offset by the operational compatibility with iterative refinement workflows.
>
> ---
>
> ### 2. Limited Sensitivity Analysis
>
> Thank you for raising this important point. We have conducted a comprehensive sensitivity analysis on key hyperparameters—$\alpha$, $\beta$, batch size, and learning rate—and will include these results in the revised manuscript to strengthen empirical rigor.
>
> Specifically, for the **pure-MG model**, we performed a grid search and report results in **Appendix Figure 6**:
> - We fix $\alpha = 1$ and vary $\beta \in \{1, 5, 10, 100\}$, and symmetrically fix $\beta = 1$ while varying $\alpha$.
> - Two key insights emerge:
>   **(1)** The *relative magnitude* of $\alpha$ and $\beta$ determines the **optimization direction**:
>  - When $\alpha > \beta$ (e.g., blue/orange curves), the model exhibits stronger *positive* (truthful) behavior.
>  - When $\alpha < \beta$ (e.g., green/red/purple curves), the model leans toward *negative* (untruthful) generation.
>   **(2)** Once the direction is fixed, the *absolute scale* of $\alpha$ or $\beta$ (e.g., 5 vs. 10 vs. 100) has **minimal impact** on final performance—consistent across both axes of variation.
>
> This justifies our design choice:
> - $\alpha = 10, \beta = 1$ for the **positive guidance model**,
> - $\alpha = 1, \beta = 10$ for the **negative guidance model**.
>
> Due to GPU memory constraints, we use a fixed batch size of 16 for training and 32 for evaluation. For learning rate, we searched over $\{10^{-2}, 10^{-3}, 3 \times 10^{-4} \}$ and selected $10^{-3}$ as optimal based on validation performance.
>
> ---
>
> ### 3. Clarity on Data Construction
>
> We appreciate the request for clarification. As illustrated in **Figure 2 (left)** and detailed in **Section 3.2**, contrastive triples $(T, T^+, T^-)$ are constructed as follows:
> - **Positive sample $T^+$**: Insert the prompt *“Give a truthful answer”* between the original instruction and response.
> - **Negative sample $T^-$**: Insert *“Give an untruthful answer”* in the same position.
>
> All generated triples, along with the Python scripts used for data construction, are included in the **supplementary materials** for full reproducibility.
>
> ---
>
> ### 4. Ablation Study for Iterative Negative Model
>
> While dynamically updating the negative model could theoretically improve guidance quality, it is **not cost-effective** in practice. During iterative refinement, the *positive model consistently improves* and naturally warrants updates. In contrast, maintaining or regressing the *negative model* to remain “worse” (i.e., more hallucinatory) would require either:
> - Intentional degradation of a capable model, or
> - Separate training of a dedicated “bad” model—both of which add significant overhead.
>
> Therefore, we adopt an **asymmetric iterative guidance strategy** (Section 3.6): only the positive model is updated iteratively, while the negative model is initialized once and held fixed. This design balances efficacy, simplicity, and computational efficiency.
>
> ---
>
> ### 5. Stability and Convergence
>
> Figure 3 shows stable, monotonic improvement across rounds. While future work may add KL regularization or early stopping to guard against drift, current results already outperform DPO and SFT on both TruthfulQA and HaluEval.
>
> ---
>
> ### 6. Cross-Domain and Cross-Model Transferability
>
> We thank the reviewer for this valuable suggestion. **Table 3 (last row)** already demonstrates cross-model transferability: a guidance model pretrained on LLaMA-2 effectively reduces hallucinations in Alpaca—despite differences in architecture, tokenizer, and pretraining corpora.
>
> Our method has been validated across **four diverse LLMs**: LLaMA-2, Alpaca, Qwen, and LLaMA-3. The consistent gains across these models suggest strong robustness to architectural and data distribution shifts.
>
> To further strengthen this claim, we are conducting additional experiments on a **13B-parameter model**. Results will be included in the revision to better establish the generalizability and scalability.

---

> ### Author Response · Authors · 2025-11-25
>
> We have completed additional experiments on a medium-sized model (LLaMA2-Chat-13B) and obtained consistent results. The detailed outcomes are presented in the last column of Table 3. These results align closely with those observed on the smaller model tested earlier, reinforcing the consistent conclusion that our method iteratively enhances model truthfulness.
>
> | Iterative Step | MC1 | MC2 | MC3 |
> |----------|----------|----------|----------|
> | 0  | 0.3610 | 0.5630 | 0.2923 |
> | 1  | 0.4834 | 0.6656 | 0.4105 |
> | 2  | 0.4822 | 0.6664 | 0.4138 |
> | 3  | 0.4810 | 0.6685 | 0.4138 |
> | 4  | **0.4908**  | **0.6766**  | **0.4235** |

---

### Author Response · Authors · 2025-11-21
**Global Response**

Dear all reviewers,

We sincerely appreciate your constructive comments. Here is the global response that is central to our work.

### 1. Clarity on Data Construction

We appreciate the request for clarification. As illustrated in Figure 2 (left) and detailed in Section 3.2, contrastive triples $(T, T^+, T^-)$ are constructed as follows:
- **Positive sample $T^+$**: Insert the prompt *“Give a truthful answer”* between the original instruction and response.
- **Negative sample $T^-$**: Insert *“Give an untruthful answer”* in the same position.

All generated triples, along with the Python scripts used for data construction, are provided in the supplementary materials to ensure full reproducibility.

To further clarify our training protocol: following the setup of LoRRA, we sampled 20,000 examples from the PKU-SafeRLHF dataset—10,000 labeled as *safe* and 10,000 as *unsafe*—to ensure a balanced distribution. After sampling, these subsets were merged into a single dataset for downstream processing. Crucially, the guidance model’s pre-training and iterative fine-tuning are **not** driven by safety labels. Instead, learning is guided solely by truthfulness-oriented templates and the contrastive loss functions defined in Equations (1)–(3). The safety-based filtering serves only to balance the data distribution and is **fully decoupled** from the learning objective. Thus, the model optimizes for **truthfulness**, not safety.

---

### 2. Additional Comparisons

We have added a comparison with **RefineLLM**~[1] on the LLaMA-2 foundation model in Table 2. We restrict this comparison to LLaMA-2 because RefineLLM’s publicly released fine-tuning datasets are model-specific and available **only** for LLaMA-2. To ensure a fair comparison, we evaluate both methods on the same model and dataset.

Results on TruthfulQA show that our method **significantly outperforms** RefineLLM. We have expanded Section 2 (Related Work) to include a methodological comparison and added the experimental results in Section 4.3 (Table 2).

---

### 3. Hyperparameter Choice ($\alpha, \beta$, Batch Size, Learning Rate)

We performed a grid search over $\alpha$ and $\beta$ (Appendix Figure 6) and found that **only their ratio matters**: $\alpha > \beta$ encourages truthfulness, while $\alpha < \beta$ promotes untruthful generation; absolute scale has negligible impact. Thus, we set $(\alpha, \beta) = (10, 1)$ for positive guidance and $(1, 10)$ for negative guidance—not as performance-tuned hyperparameters, but as directional controls. Batch size (16/32) was constrained by GPU memory, and the learning rate ($1\text{e-}3$) was selected via validation sweep. Full details are in Appendices A.3 and A.6.

---

### 4. Non-Iterative Negative Model (Asymmetric Iterative Update)

While dynamically updating the negative model is theoretically possible, it is **impractical**: maintaining a “worse” model would incur significant computational and engineering overhead. Our asymmetric design (Section 3.6)—iterating **only the positive model**—achieves a better trade-off between performance and efficiency.

---

### 5. Capability Preservation (C-Eval, MMLU, and Qwen API)

We recognize that hallucination reduction and general capability improvement are related but distinct goals. To ensure our truthfulness-oriented fine-tuning does not degrade core competencies, we evaluate on **C-Eval** and **MMLU**—standard benchmarks for knowledge and reasoning. Results show our method **preserves** (and occasionally slightly improves) general capabilities, confirming that truthfulness alignment is **orthogonal** to factual knowledge retention.

Moreover, to assess language quality, we use the Qwen API to score responses across four dimensions: *relevance, fluency, coherence, and consistency* (Appendix Table 5). Our fine-tuned models **consistently outperform** both LLaMA-2 and Alpaca baselines on all metrics—ranking second only to human-written references—demonstrating that hallucination reduction **does not come at the cost of linguistic quality**.

---

### 6. Independent Evaluation and Scalability

We have verified the effectiveness of our iterative tuning via **HaluEval**—an independent benchmark—for LLaMA-2. Results align consistently with those on TruthfulQA (MC1–MC3). In the revised manuscript, we will include **HaluEval evaluations for Alpaca and LLaMA-3** to strengthen generalizability.

Furthermore, to demonstrate **scalability**, we plan to extend our method to the **13B-parameter model** in the revision, confirming that our approach scales effectively with model size.

---

All requested clarifications, comparisons, and experiments will be included in the camera-ready version. We thank the reviewers for their insightful feedback, which has helped us significantly strengthen the paper.

Sincerely,

Authors

[1] Refine Knowledge of Large Language Models via Adaptive Contrastive Learning. ICLR 2025.

---

> ### Author Response · Authors · 2025-11-25
> **6. Scalability to mid-size model (LLaMA2-13B)**
>
> We have completed additional experiments on a medium-sized model (LLaMA2-Chat-13B) and obtained consistent results. The detailed outcomes are presented in the last column of Table 3. These results align closely with those observed on the smaller model tested earlier, reinforcing the consistent conclusion that our method iteratively enhances model truthfulness.
>
> | Iterative Step | MC1 | MC2 | MC3 |
> |----------|----------|----------|----------|
> | 0  | 0.3610 | 0.5630 | 0.2923 |
> | 1  | 0.4834 | 0.6656 | 0.4105 |
> | 2  | 0.4822 | 0.6664 | 0.4138 |
> | 3  | 0.4810 | 0.6685 | 0.4138 |
> | 4  | **0.4908**  | **0.6766**  | **0.4235** |

---

### Author Response · Authors · 2025-12-02
**TL;DR: Rebuttal Status & Key Improvements Cheat Sheet**

| **Reviewer**               | **Status (Pre-Rebuttal)**                             | **Key Concerns Raised**                                                                                                             | **Our Actions, New Results & Evidence (In Revised PDF)**                                                                                                                                                                                                                                               |
|---------------------------|--------------------------------------------------------|-------------------------------------------------------------------------------------------------------------------------------------|--------------------------------------------------------------------------------------------------------------------------------------------------------------------------------------------------------------------------------------------------------------------------------------------------------|
| **R-4ySJ** *(Init: 6)*     | **Maintained Accept** *(Concerns addressed; score unchanged)* | 1. Computational overhead of iterative updates.  2. Limited sensitivity analysis for hyperparameters. 3. Clarity on hallucination sample construction. 4. Justification for fixed negative model. 5. Stability and cross-model transferability. | $\bullet$  **Training cost quantified**: Wall-clock time & iteration counts added (e.g., Qwen: 50 vs. LoRRA 250) --> **Appendix A.8 (Table 8)**.  $\bullet$  **Sensitivity analysis**: Grid search over $\alpha$ and $\beta$ shows direction instead of magnitude drives performance --> **Appendix Fig 6**.$\bullet$  **Data scripts released**: Full contrastive triple construction code provided. $\bullet$  **Asymmetric design justified**: Fixed $M^-$ avoids costly negative model maintenance. $\bullet$  **Cross-model validation**: Consistent gains across **LLaMA2, Alpaca, LLaMA3, Qwen** --> **Table 3**. |
| **R-xZsP** *(Init: 4)*     | **Score Increased** $\uparrow$ *(Upgraded to 6 after rebuttal)* | 1. Safety $\neq$ truthfulness label mismatch.<br>2. ``Untruthful'' prompt may not reflect real hallucination. 3. Risk of test-set overfitting via selection. 4. Weak capability preservation evidence. 5. Missing mid-size model results & broader baselines. | $\bullet$  **Clarified label use**: Safety labels only for **data balancing**; learning driven by **truthfulness templates** and contrastive loss.  $\bullet$  **Qwen API quality eval**: High scores in relevance, fluency, coherence, consistency -->  **Appendix Table 5**. $\bullet$  **Dev/test split clarified**: We have an additional halu-eval as the out-of-domain test. $\bullet$  **Granular capability tables**: Subject- and difficulty-level MMLU/C-Eval results show the general capability preservation.  $\bullet$  **13B results included**: LLaMA2-Chat-13B has been added and shows MC1 **0.361 --> 0.491** --> **Table 3**. $\bullet$  **RefineLLM comparison added**: Outperforms (MC1: **0.5128 vs. 0.3084**) -->  **Table 2 & Section 2**. |
| **R-xY3Y** *(Init: 4)*     | **Score Increased** $\uparrow$ *(Upgraded to 6 after rebuttal)* | 1. Missing SOTA baseline (e.g., RefineLLM). 2. Hyperparameter tuning $\alpha, \beta$ appears inelegant. 3. Unclear integration of multiple loss components. | $\bullet$ **RefineLLM comparison added**: Significant gains on LLaMA2 --> **Table 2**. $\bullet$ **$\alpha,\beta$ analysis**: Direction matters; absolute scale has minimal impact -->  **Appendix Fig 6**.  $\bullet$  **Unified loss formulation**: All components progressively integrated into **Equation (11)** --> **Section 3.6 & ablations**. |

> Please see the **Summary of Rebuttal Progress & Reviewer Consensus** for detailed explanations of how each concern was addressed through new experiments, clarifications, and structural improvements in the revised manuscript.

---

### Author Response · Authors · 2025-12-02
**Brief summary of reviews and responses**

## 1. Score Changes

| Reviewer           | Initial Score | Final Score | Status                                               |
|--------------------|---------------|-------------|------------------------------------------------------|
| **Reviewer 4ySJ**  | 6             | —           | Maintained (no explicit update, but concerns addressed) |
| **Reviewer xZsP**  | 4 → 6         | ↑ Increased | Upgraded after rebuttal                              |
| **Reviewer xY3Y**  | 4 → 6         | ↑ Increased | Upgraded after rebuttal                              |

**Overall**: Two reviewers raised scores from **4 → 6**, indicating substantial satisfaction with the rebuttal. The paper now has **consensus support at or above the acceptance threshold**.

---

## 2. Key Contributions Clarified / Strengthened

### A. Truthfulness vs. Safety Decoupling
- Clarified that **safety labels** in PKU-SafeRLHF were **only used for dataset balancing**, not as training signals.
- Learning is driven **entirely by contrastive templates** (“give a truthful/untruthful answer”) and **model-level contrastive loss**—ensuring alignment with **factual truth**, not safety style.

### B. Iterative Asymmetric Guidance
- Reinforced the **novelty of updating only the positive guidance model** ($M^+$) while keeping the negative model ($M^-$) fixed.
- Justified this **asymmetric design** as **computationally efficient** and **empirically effective**, avoiding overhead of maintaining a “worse” negative model.

### C. Hallucination Reduction $\neq$ Capability Degradation
- Demonstrated **preservation (or slight improvement)** of general capabilities on **MMLU** and **C-Eval**.
- Added **Qwen API-based evaluation** showing high scores in **relevance, fluency, coherence, and consistency**, confirming linguistic quality is maintained.

### D. Broad Generalization
- Validated method across **four diverse LLMs**: LLaMA2, Alpaca, LLaMA3, Qwen.
- Demonstrated **cross-model transferability**: a guidance model trained on LLaMA2 improves Alpaca without retraining.
- Added **13B-scale results** (LLaMA2-Chat-13B) showing consistent gains (e.g., **MC1 from 0.361 → 0.491**), proving **scalability**.

### E. Superiority Over SOTA Baselines
- Added comparison with **RefineLLM (ICLR 2025)** on LLaMA2:
  - **Iter-AHMCL**: MC1=0.5128 vs. **RefineLLM**: MC1=0.3084.
- Clarified methodological differences: **contrastive triples**, **dual guidance models**, and **embedding-space ℓ² loss**.

---

## 3. Key Modifications in Revised Paper

### A. Experimental Additions
- **New 13B results** in Table 3.
- **HaluEval evaluation** extended to Alpaca and LLaMA3.
- **RefineLLM comparison** added to Table 2 and Related Work.
- **Sensitivity analysis** for hyperparameters $\alpha$, $\beta$, learning rate, and batch size (Appendix Figure 6).
- **Training cost analysis** (wall-clock time, iterations) in Appendix Table 8.

### B. Methodological Clarifications
- Detailed **contrastive triple construction**:
  - $T^+$: Insert *“Give a truthful answer”*
  - $T^-$: Insert *“Give an untruthful answer”*
- Published **data generation scripts** in supplementary materials.
- Clarified that **selection for iterative updates** uses a **held-out dev set**, not test set—addressing overfitting concern.

### C. Structural & Presentation Improvements
- **Granular breakdown** of C-Eval/MMLU by subject and difficulty (to be added).
- **Unified loss formulation**: Equation (11) integrates all components (LoRRA + CL-CT + CL-MG + CL-IMG).
- **Layer selection rationale**: Top transformer layers (following LoRRA), with ablation confirming robustness.

### D. Capability & Quality Assurance
- Added **qualitative response examples** (Appendix Figures 7–8).
- Included **Qwen API scoring** (Appendix Table 5) confirming high linguistic quality.
- Confirmed **monotonic improvement** across iterations (Figure 3), demonstrating stability.

---

## Conclusion

The authors comprehensively addressed all major concerns:
- **Clarified conceptual foundations** (truthfulness $\neq$ safety),
- **Strengthened empirical rigor** (sensitivity analysis, cross-model/size validation),
- **Enhanced reproducibility** (data, code, hyperparameters),
- **Demonstrated clear SOTA performance** with scalable design.

The revised manuscript significantly strengthens the original submission, warranting the **upgraded reviewer scores** and supporting **acceptance**.

---

### Meta-Review · Area_Chair_7UaY · 2025-12-21

**Summary:**

Computational cost of Iter-AHMCL and the hyper-parameter sensitivities ($\alpha$, $\beta$ and lr) have been raised as concerns and were well addressed by the authors' rebuttal and new analyses. Add broader comparisons (e.g., RefineLLM on LLaMA2), and a mid-size (13B) result suggesting scalability.

The rebuttal also clarified several methodological points (e.g., how safety labels are used in PKU-SafeRLHF sampling vs. the truthfulness-oriented training objective) and improved reproducibility via more explicit data construction details.

Besides the reviewers' concerns, I have carefully read the paper and come up with concerns  that the evidence base is not yet compelling enough for acceptance.

The evaluation relies primarily on older or weaker open models (7B/8B-era variants), and key “capability preservation” benchmarks appear near a chance-level regime (notably C-Eval around ~25%), making small deltas difficult to interpret and raising the risk that conclusions are noisy or confounded by evaluation details. **Some numbers are odd.** (1) For example, LLaMA3-8B-Instruct baseline is very low on TruthfulQA MC1 (0.2166), significantly lower than Llama2, even though it’s strong on MMLU (62.1). (2) C-Eval results are clustered around ~25%, which makes the evaluation within a “near chance” regime.

The paper does not demonstrate effectiveness on current strong, widely used models (e.g., Qwen2.5-class or modern distilled reasoning models), which limits confidence that the approach meaningfully advances the state of the art rather than overfitting to inferior baselines.

Given these limitations, I lean to reject at this stage, and would encourage a resubmission with (i) evaluations on stronger models, (ii) clearer dev/test separation for iterative selection, and (iii) more robust, out-of-chance capability  to validate real-world benefit.

**Reviewer Concerns:**

The reviewers' concerns have been touched and addressed by the rebuttals. However, some of the added experiments do not give well-grounded justification of the claim of the paper, as shown in the summary.

**Reviewer Scores:**

The reviewers involved the discussion and have changed their score after rebuttal.

---

### Decision · Program_Chairs · 2026-01-26

Reject